# Synthesis of Ni_3_S_2_ and MOF-Derived Ni(OH)_2_ Composite Electrode Materials on Ni Foam for High-Performance Supercapacitors

**DOI:** 10.3390/nano13030493

**Published:** 2023-01-26

**Authors:** Meng Shao, Jun Li, Jing Li, Yanan Yan, Ruoliu Li

**Affiliations:** School of Materials Science and Engineering, Shanghai University of Engineering Science, Shanghai 201620, China

**Keywords:** Ni(OH)_2_/Ni_3_S_2_/NF, honeycomb-like structure, two-step hydrothermal, alkalization, MOF-derived Ni(OH)_2_, supercapacitor

## Abstract

Honeycomb-like Ni(OH)_2_/Ni_3_S_2_/Ni foam (NF) was fabricated via a two-step hydrothermal process and subsequent alkalization. Ni_3_S_2_ with a honeycombed structure was in-situ synthesized on the NF surface by a hydrothermal process. MOF-derived Ni(OH)_2_ nanosheets were then successfully grown on the Ni_3_S_2_/NF surface by a second hydrothermal process and alkaline treatment, and a large number of nanosheets were interconnected to form a typical honeycomb-like structure with a large specific surface area and porosity. As a binder-free electrode, the prepared honeycomb-like Ni(OH)_2_/Ni_3_S_2_/NF exhibited a high specific capacitance (2207 F·g^−1^ at 1 A·g^−1^, 1929.7 F·g^−1^ at 5 mV·s^−1^) and a remarkable rate capability and cycling stability, with 62.3% of the initial value (1 A·g^−1^) retained at 10 A·g^−1^ and 90.4% of the initial value (first circle at 50 mV·s^−1^) retained after 5000 cycles. A hybrid supercapacitor (HSC) was assembled with Ni(OH)_2_/Ni_3_S_2_/NF as the positive electrode and activated carbon (AC) as the negative electrode and exhibited an outstanding energy density of 24.5 Wh·kg^−1^ at the power density of 375 W·kg^−1^. These encouraging results render the electrode a potential candidate for energy storage.

## 1. Introduction

Owing to economic development and population growth, the world’s demand for energy is growing. However, traditional energy sources present some disadvantages, such as limited resources and environmental unfriendliness [1,2,3,4]. Therefore, ecofriendly energy resources and sustainable electrical energy storage devices (especially lithium-ion batteries and supercapacitors) must be urgently developed. Compared with batteries, supercapacitors have the advantages of high charge energy storage, fast charge and discharge, long-term stability as well as environmental friendliness [5,6,7], and have become the research focus in recent years. Depending on their charge storage mechanism, supercapacitors are divided into electric double layer capacitors (EDLCs) and pseudo-capacitors [8]. Compared to EDLCs, pseudo-capacitors have a higher specific capacity owing to their quick and reversible Faraday reactions [9]. The electrode material is the most critical factor affecting the performance of supercapacitors [2,6,10] and is mainly classified as carbon materials, conductive polymers, and transition metal compounds [11]. Transition metal compounds show superior specific capacity due to their abundant oxidation–reduction states and small internal resistance [12]. Currently, the widely studied transition metal compounds mainly include oxides (NiO and MnO_2_), sulfides (Ni_3_S_2_ and MoS_2_), and hydroxides (Ni(OH)_2_ and Co(OH)_2_). An S atom with lower electronegativity than an O atom is beneficial to electron transportation and could further promote the redox reactions, resulting in transition metal sulfides exhibiting higher specific capacities than oxides [13].

Among transition metal sulfides, Ni_3_S_2_ is a promising electrode material due to its abundant resources and high theoretical specific capacitance and electrical conductivity [11,14,15]. However, one drawback of Ni_3_S_2_ is its poor cycling stability due to the fall-off of the active materials caused by drastic volume changes during charge–discharge [16]. Huo et al. [17] designed 3D Ni_3_S_2_ nanosheet arrays grown on Ni foam (NF) by a hydrothermal process. The electrode showed an excellent specific capacitance of 1370 F·g^−1^ at 2 A·g^−1^ and a remarkable rate capability (952.0 F·g^−1^ at 20 A·g^−1^); however, its specific capacitance only remained at 91.4% after 1000 cycles. Chen et al. [18] prepared Ni_3_S_2_ with a mixed structure of flakes and granules using the hydrothermal method. The product showed a specific capacitance of 736.64 F·g^−1^ at 0.8 A·g^−1^ and maintained 82% of its original specific capacitance after 1000 cycles. An effective solution is to combine Ni_3_S_2_ with other materials to form a composite. In the absence of a direct contact between Ni_3_S_2_ and electrolyte, the internal electrode material can be effectively prevented from falling off under the continuous influence of high current, and the cycling stability is consequently improved. At present, the electrode materials for Ni_3_S_2_ composite mainly include carbon materials [19], polymer compounds [20], MXenes [21], sulfides [22], oxides [23], and hydroxides [24,25,26]. Among these, hydroxides have attracted attention, especially for Ni(OH)_2_ with a high theoretical specific capacity and abundant resources [27]. Composite electrodes containing Ni(OH)_2_ and Ni_3_S_2_ can effectively relieve the detachment of the internal material (Ni_3_S_2_) from the current collector, resulting in an improvement in the cycling stability. They can also provide additional active sites from the external material (Ni(OH)_2_), leading to a high charge storage. Wang et al. [26] fabricated a NF-Ni_3_S_2_@Ni(OH)_2_-graphene sandwich structure by a two-step hydrothermal method. Ni(OH)_2_ was in the middle of the structure, with rGO and Ni_3_S_2_ attached above and below, respectively. Owing to the structural feature, the electrode exhibited a high specific capacitance of 2258 F·g^−1^ at 1 A·g^−1^ and excellent cycling stability with no capacitance reduction after 3000 cycles. Pan et al. [24] additionally fabricated Ni_3_S_2_ nanorods in situ on a NF and then loaded them with the ultrathin Ni(OH)_2_ nanosheets by a two-step hydrothermal process. The electrode exhibited remarkable specific capacitance of 826 C·g^−1^ at 1 A·g^−1^, excellent rate capability (50% retention at 20 A·g^−1^), and outstanding cycling stability (82% of its initial capacitance retained over 10,000 cycles). For further improvement in cycling stability and specific capacity, an effective method is to construct a Ni(OH)_2_/Ni_3_S_2_ composite electrode with multiple channels and a high specific surface area by changing the structure of Ni(OH)_2_. The abundant pores can provide a large number of diffusion channels, greatly shortening the diffusion path of ions and reducing the stability decline caused by volume strain [28]. Meanwhile, the high specific surface area can provide additional active sites and increase the charge storage of electrode materials [29]. The metal–organic framework (MOF), which has a high porosity and surface area, is often used as a precursor and template to derive Ni(OH)_2_ [29,30,31]. However, a suitable strategy to compound Ni(OH)_2_ derived from Ni-MOF with Ni_3_S_2_ has not been discovered.

In this work, honeycomb-like Ni(OH)_2_/Ni_3_S_2_/NF (Ni(OH)_2_ derived from Ni-MOF) grown on a NF was synthesized by a two-step hydrothermal method coupled with an alkalization treatment. Highly conductive Ni_3_S_2_ as a carrier was fabricated in situ on NF by the first hydrothermal method without using any organic binder, and Ni(OH)_2_ nanosheets derived from Ni-MOF were then incorporated as a support by the second hydrothermal method and alkalization treatment in 6 M KOH for 6 h at 75 °C. The change in the morphology structure from Ni_3_S_2_/NF to Ni(OH)_2_/Ni_3_S_2_/NF was investigated, and the relationship between this change and the electrochemical performance was analyzed. When tested in three- and two-electrode systems, the as-prepared honeycomb-like Ni(OH)_2_/Ni_3_S_2_/NF electrode demonstrated excellent electrochemical performance.

## 2. Experimental methods

### 2.1. Materials

NF (thickness of 1 mm, Guangjiayuan New Material Co., Kunshan, China), thiourea (≥99%, Shanghai Titan Scientific Co., Ltd., Shanghai, China), nickel nitrate hexahydrate (Ni(NO_3_)_2_·6H_2_O, 98.5%, Shanghai Titan Scientific Co., Ltd., Shanghai, China), 1,3,5-benzenetricarboxylic acid (PTA, 99%, Aladdin Industrial Corporation, Shanghai, China), and N,N-dimethylformamide (DMF, ≥99.9%, Aladdin Industrial Corporation) were used without further operation.

### 2.2. Synthesis of Ni_3_S_2_ and MOF-Derived Ni(OH)_2_

Prior to the experiments, the NF (1.5 × 2 cm) was successively cleaned with acetone, 1 M HCl aqueous solution, deionized water (DI), and absolute ethanol in an ultrasonic cleaner to remove the organic matter and oxide layer and then dried in a vacuum drying oven at 60 °C for 12 h. The cleaned grey NF was placed in a 50 mL Teflon-lined stainless autoclave containing a homogeneous solution of 35 mL of DI and 10 mmol thiourea. The Teflon stainless autoclave was kept at 120 °C for 5 h to synthesize Ni_3_S_2_, which was subsequently cooled to room temperature. The samples were taken out and gently cleaned in DI and absolute ethanol. In brief, 2 mmol Ni(NO_3_)_2_·6H_2_O and 0.5 mmol PTA were dissolved in 30 mL of DMF and stirred for 60 min. The NF with in-situ synthesized Ni_3_S_2_ and the above solution were transferred into a 50 mL Teflon stainless autoclave to heat at 120 °C for 9 h, after which Ni-MOF was deposited onto Ni_3_S_2_. The resultant sample was gently cleaned several times by DMF and DI and then immersed in KOH solution at 75 °C for 6 h to transform Ni-MOF into MOF-derived Ni(OH)_2_. The sample was washed in DI before vacuum drying 60 °C for 12 h to obtain NF coated with green materials. The above synthesis process is clearly and intuitively illustrated in Figure 1.

### 2.3. Material Characterization

X-ray diffraction patterns were obtained to analyze the phase constituents of the resultant products on an X-ray diffractometer (XRD, Bruker, Karlsruhe, Germany) with Cu-Kα radiation (γ = 0.1540560 nm). Their morphologies were observed by a field-emission scanning electron microscope (SEM, S-4800, Hitachi, Tokyo, Japan) and a transmission electron microscope (TEM, JEM-2100F, JEOL, Tokyo, Japan) equipped with an energy dispersive spectrometer (EDS, X-MAX 65T, OXFORD, UK). Their chemical compositions and chemical valences were analyzed with an X-ray photoelectron spectroscope (XPS, Thermo Fisher Scientific, Waltham, MA, USA).

### 2.4. Electrochemical Tests

#### 2.4.1. Three-Electrode System

The electrochemical performance was tested by cyclic voltammetry (CV), galvanostatic charge-discharge tests (GCD), and electrochemical impedance spectroscopy (EIS) in a three-electrode system. CV and EIS were tested on a CS350H electrochemical workstation (CORRTEST, Wuhan, China), and the GCD was tested on a CHI 760E electrochemical workstation (CH Instrument Inc, Shanghai, China). The NF sheet loaded with Ni_3_S_2_/MOF-derived Ni(OH)_2_ and Ni_3_S_2_ and the naked NF sheet free of active substances were used as the working electrode, and a saturated calomel electrode and a graphite sheet acted as the reference and counter electrodes, respectively. A 6 M KOH aqueous solution was chosen as the electrolyte. Prior to the tests, the prepared working electrode was soaked in 6 M KOH solution for 12 h to ensure full contact between the active substance and the electrolyte. CV measurements were performed within a potential window of −0.1 V to 0.6 V at the scan rate ranging from 5 mV·s^−1^ to 50 mV·s^−1^. GCD tests were carried out in the potential window from 0 to 0.4 V under different current densities (1, 2, 4, 6, 8, and 10 A·g^−1^). CV tests to measure the cycling stability of the electrode (the NF sheet loading Ni_3_S_2_/MOF-derived Ni(OH)_2_) were conducted for 5000 cycles at 50 mV·s^−1^. An EIS test was carried out in the frequency range of 10^5^ Hz to 0.01 Hz. 

The specific capacity (*C_α_* and *C_m_*, F·g^−1^) of electrode materials was calculated from the CV and GCD curves using Formulas (1) and (2), respectively [28,29]:

For the CV tests:(1)$$C_{\alpha }=\frac{\int IdV}{m\times v\times \Delta V}$$

For the GCD tests:(2)$$C_{m}=\frac{I\;\times \Delta t}{m\times \Delta V}$$
where *I* (A) is to the current, *m* (g) is the mass of the working electrodes, *v* (mV·s^−1^) is the scan rate, *∆V* (V) is the potential range, and *∆t* (s) is the discharge time.

#### 2.4.2. Two-Electrode System

A hybrid supercapacitor (HSC) was assembled by using an Ni_3_S_2_/MOF-derived Ni(OH)_2_ electrode as the positive electrode, activated carbon as the negative electrode, filter paper as the separator, and 6 M KOH as the electrolyte. For the preparation of the negative electrode, 90 wt.% activated carbon (AC) and 10 wt.% poly vinylidene fluoride were mixed in an appropriate amount of N-methyl-2-pyrrolidone solution and then coated on a NF. The coated NF was then folded, kept at a pressure of 10 MPa for 10s, and dried at 120 °C for 6 h. The mass ratio of the NF sheet loaded with Ni_3_S_2_/MOF-derived Ni(OH)_2_ as the positive electrode and AC as the negative electrode was calculated by the following charge balance formula (3) [32]:(3)$$\frac{m^{-}}{m^{+}}=\frac{C_{\alpha }^{+}\times \Delta V^{+}}{C_{\alpha }^{-}\times \Delta V^{-}}$$
where *m* (g) is the active material mass, *C_α_* (F·g^−1^) is the specific capacity under the CV tests, and *∆V* (V) is the potential window for positive electrodes (+) and negative electrodes (−). On the basis of the charge balance formula, the mass ratio of positive electrodes and negative electrodes was calculated to be approximately 0.2.

In the two-electrode system, the electrochemical workstation used to measure CV and GCD was the same as that in the three-electrode system. The specific capacity (*C_device_*, F·g^−1^) of the HSC device was obtained by the following formula (4) [33]:(4)$$C_{device}=\frac{I\times \Delta t}{m_{total}\times \Delta V}$$
where *I* (A) is the constant discharge current, *m_total_* (g) is the sum of the masses of the positive and negative active materials, *∆V* (V) is the potential window, and *∆t* (s) is the discharge time.

The energy density (*E*, Wh·kg^−1^) and power density (*P*, W·kg^−1^) of the HSC device were computed according to Formulas (5) and (6), respectively [34]:(5)$$E=\frac{C_{device}\times \Delta V^{2}}{2\times 3.6}$$
(6)$$p=\frac{E\times 3600}{\Delta t}$$
where *C_device_* (F·g^−1^) is the specific capacitance of the HSC device, and *∆V* (V) and *∆t* (s) are the voltage window and discharge time, respectively.

## 3. Results and Discussion

### 3.1. Structural Characterization

The as-prepared electrode was characterized by XRD to investigate its phase composition (Figure 2). Strong diffraction peaks marked with “♧” were the characteristic peaks of the substrate (NF) (JCPDS, No.01-070-0989) found at 44.6°, 51.9°, and 76.6°, corresponding to the (111), (200), and (220) planes, respectively. The diffraction peaks (marked with “♢”) at 21.8°, 31.2°, 37.9°, 44.5°, 49.8°, 50.3° 54.7°, 55.3°, and 72.6° were assigned to the (100), (1−10), (111), (200), (210), (2−10), (211), (2−11), and (310) planes of Ni_3_S_2_, respectively (JCPDS, No:01-073-0698). The sulfidization of NF can be described as follows [11]:(7)$${\left(NH_{2}\right)}_{2}CS+2H_{2}O=H_{2}S+CO_{2}+2NH_{3}$$
(8)$$2H_{2}S+3Ni=Ni_{3}S_{2}+2H_{2}$$

In addition to the diffraction peaks of NF and Ni_3_S_2_, other diffraction peaks (marked with “♤”) were detected at 19.3°, 33.1°, 38.5°, 52.1°, 59.1°, 62.7°, and 72.6°, corresponding to the (001), (100), (101), (102), (110), (111), and (201) planes of Ni(OH)_2_, respectively (JCPDS, No:00-014-0117). Yang et al. [35] failed to observe the characteristic peaks of Ni-MOF (at 15.7°, 18.4°, and 23.7°), indicating that all Ni-MOFs were completely converted to Ni(OH)_2_. The diffraction peaks related to Ni_3_S_2_ and Ni(OH)_2_ were comparatively weak due to their lower content compared with that of the NF substrate. Overall, Ni_3_S_2_ and Ni(OH)_2_ were successfully synthesized on NF.

The XPS spectra of the Ni(OH)_2_/Ni_3_S_2_/NF electrode were tested to determine the element compositions and their valences. Ni, O, and S were found to be involved in the electrode (Figure 3a). Six peaks can be well fitted in the Ni 2p high-resolution spectrum (Figure 3b). Two peaks with the binding energies of 855.7 and 873.4 eV were related to Ni^2+^, two peaks with the binding energies of 857.6 and 875.2 eV were correlated with Ni^3+^ [36], and two peaks at 861.8 and 879.7 eV belonged to the satellite peaks [1]. These results confirmed the successful synthesis of Ni_3_S_2_. For the O 1s spectrum, the peak at 530.9 eV implied the existence of OH¯ in Ni(OH)_2_ derived from Ni-MOF [29]. The peak at 532.8 eV may be attributed to the oxygen atoms adsorbed on the surface of the sample (Figure 3c) [24,37]. As shown in Figure 3d, the S 2p spectrum was composed of two peaks of S 2p_3/2_ and S 2p_1/2_ at 162.4 and 163.6 eV, respectively [38]. The other peak at 168.0 eV was the satellite peak [14]. All these findings were consistent with the XRD analysis, further proving the existence of Ni(OH)_2_ and Ni_3_S_2_.

### 3.2. Morphology Characterization

The morphology of pure NF, Ni_3_S_2_/NF, and Ni(OH)_2_/Ni_3_S_2_/NF were characterized using a field-emission scanning electron microscope. A large number of equiaxed grains with a size of approximate were observed on the skeleton of NF (Figure 4a,b). The low-magnification images of the Ni_3_S_2_/NF electrode showed that the NF skeleton was uniformly covered with a layer of dense substances (Figure 4c,d). Therefore, Ni_3_S_2_ with a honeycombed structure was in-situ synthesized on the NF surface. Ni_3_S_2_ with a similar structure was also reported by Wei et al. [39]. When the electrode was subjected to the second hydrothermal process and subsequent alkaline treatment, its surface became rough due to the abundant nanosheets protruding from the Ni_3_S_2_/NF surface (Figure 4e). Some even clustered together to form a flower-like structure, which connected to the individual nanosheets to create a typical honeycombed structure (Figure 4f). This finding indicated that the Ni(OH)_2_ derived from Ni-MOF recombined on the Ni_3_S_2_/NF surface. The honeycomb structure of Ni(OH)_2_/Ni_3_S_2_ was formed through two processes. First, Ni-MOF recombined on the surface of Ni_3_S_2_/NF by ”selective attachment“ and ”self-assembly“ [40]. Ni^2+^ and carboxyl organic ligands selectively attached to the high energy sites (protrusion) of Ni_3_S_2_ nanosheets and then self-assembled to form Ni-MOF on Ni_3_S_2_. Second, the organic ligands of Ni-MOF were replaced by OH^−^ and then converted into Ni(OH)_2_. The evolution mechanism of Ni-MOF converted to Ni(OH)_2_ is related to a “crystal–crystal conversion” mechanism [29]. When Ni-MOF was immersed in an alkaline solution, Ni(OH)_2_ with a hexagonal structure was formed due to the carboxyl organic ligands being replaced by OH^−^. The Ni(OH)_2_ molecules lined up along the crystal plane parallel to the Z-axis and grew along the X–Y crystal plane, eventually forming the sheets. A large number of ultrathin nanosheets joined to one another to form a honeycomb structure. Owing to the high porosity of Ni-MOF retained in the Ni(OH)_2_ derived from Ni-MOF and the large specific surface area involved in the honeycombed structure, the diffusion path of OH^-^ in the electrolyte was greatly shortened and additional active sites were provided. Hence, the electrochemical performance of the electrode material was improved.

Figure 5 displays the microstructure characteristics of the as-prepared samples. The sheet-like morphology of Ni_3_S_2_ can be vaguely distinguished due to discontinuous distribution and unclear boundaries (Figure 5a). The lattice fringe space was confirmed as 0.41 nm in the HRTEM image (Figure 5b), corresponding to the (100) plane of Ni_3_S_2_. After the second hydrothermal process and alkaline treatment, the sheet-like structure became evident, indicating that the Ni(OH)_2_ nanosheets derived from Ni-MOF had tightly attached to the surfaces of Ni_3_S_2_/NF nanosheets (Figure 5c). The dark part was the Ni_3_S_2_ nanosheets, around which a bright outer layer of Ni(OH)_2_ nanosheets can be detected (Figure 5d). The lattice stripes with d-values of 0.41 and 0.23 nm appeared in the inner and outer layers, respectively, which were related to the (100) plane of Ni_3_S_2_ and (101) plane of Ni(OH)_2_, respectively.

In Figure 6, the EDS element mapping analysis of Ni(OH)_2_/Ni_3_S_2_/NF revealed the co-existence of Ni, S, and O elements. This finding was consistent with the XPS results. The elements were uniformly distributed on the nanosheets. The content of O and Ni elements was higher than that of the S element, possibly because Ni_3_S_2_ was coated by Ni(OH)_2_. This result further proved the successful preparation of Ni_3_S_2_/Ni(OH)_2_/NF.

### 3.3. Electrochemical Tests

The electrochemical performance of all electrodes was investigated in a 6 M KOH aqueous solution with a standard three-electrode system. Figure 7a shows the CV curves of NF, Ni_3_S_2_/NF, and Ni(OH)_2_/Ni_3_S_2_/NF at the scanning rate of 10 mV·s^−1^ in a potential of −0.1 V to 0.6 V. For the NF electrode, the curve during the positive sweeping almost coincided with that during the negative sweeping and no redox peaks were observed, implying that NF as an inert substance hardly has the ability to store electrical charges. When Ni_3_S_2_ was in-situ synthesized on the NF, the CV curve seriously deviated from the rectangle of EDLCs and showed distinct redox peaks at 0.29 and 0.10 V, demonstrating the typical pseudo-capacitance behavior of the electrode [23,39]. The Faradic redox reaction can be described as follows [7,41]:(9)$$Ni_{3}S_{2}+3\;OH^{-}\;↔\;Ni_{3}S_{2}{\left(OH\right)}_{2}+3\;e^{-}$$

After Ni_3_S_2_ was covered with the Ni(OH)_2_ nanosheets, the area surrounding the CV curve became further enlarged. Clear inspection revealed that the oxidation and reduction peaks widened. Other redox peaks were discerned at 0.32 and 0.08 V, corresponding to the following reactions [7,41]:(10)$$Ni{\left(OH\right)}_{2}+OH^{-}\;↔\;NiOOH+H_{2}O+e^{-}$$

The area surrounding the CV curves represents the number of charges involved in the Faradic redox reaction [42]. Ni(OH)_2_/Ni_3_S_2_/NF had the largest surrounding area, followed by Ni_3_S_2_/NF. This finding indicated that the integration of Ni(OH)_2_ into Ni_3_S_2_ contributes to improving the charge storage capacity. As calculated by Formula (1), the specific capacitances of NF, Ni_3_S_2_/NF, and Ni(OH)_2_/Ni_3_S_2_/NF were 3.3, 623.8, and 1929.7 F·g^−1^, respectively. The Ni(OH)_2_ derived from Ni-MOF with a specific surface area showed an increase in specific capacitance by approximately 209%. The high charge storage of Ni(OH)_2_/Ni_3_S_2_/NF is closely related to the high specific capacitance (2082 F·g^−1^) of Ni(OH)_2_ and its unique morphology. On the one hand, XRD and XPS results showed that the Ni-MOF-derived Ni(OH)_2_ was successfully synthesized after the second hydrothermal reaction and subsequent alkalization treatment. The attached Ni(OH)_2_ with a layered structure is conducive to ion transport and therefore has a high specific capacity. As shown in the SEM image, the Ni(OH)_2_/Ni_3_S_2_/NF surface became rough after recombination, forming a honeycomb structure with a large contact area with electrolyte and thus providing additional active sites for the Faradic redox reaction. On the other hand, the Ni(OH)_2_ derived from Ni-MOF with a high porosity allowed OH^−^ ions to readily penetrate the interior of the electrode material and interact with Ni_3_S_2_/NF. The active substance of Ni_3_S_2_ was developed directly onto the NF substrate, eliminating the interface resistance between the two and enhancing the electron transport rate. These factors led to a synergistic effect, which can remarkably increase the specific capacitance of electrode materials. 

Figure 7b displays the CV curves of Ni(OH)_2_/Ni_3_S_2_/NF samples at the scanning rates ranging from 5 mV·s^−1^ to 50 mV·s^−1^. All CV curves showed good symmetry, indicating that the samples have excellent electrochemical reaction reversibility. The specific capacitance of Ni(OH)_2_/Ni_3_S_2_/NF at different scanning rates was 1977.1 (5 mV·s^−1^), 1929.7 (10 mV·s^−1^), 1858.0 (15 mV·s^−1^), 1582.3 (25 mV·s^−1^), and 1063.5 F·g^−1^ (50 mV·s^−1^). When the scanning rate was increased from 5 mV·s^−1^ to 50 mV·s^−1^, the specific capacitance still remained at 53.8% of that obtained at 5 mV·s^−1^, demonstrating the remarkable rate capability of Ni(OH)_2_/Ni_3_S_2_/NF. With the gradual increase in the scanning rate, the specific capacitance showed a downward trend (Figure 7c) because the reaction was controlled by diffusion. At a high scanning rate, OH^−^ ions cannot diffuse to the electrode surface in time, resulting in a decrease in specific capacity. Moreover, the position of oxidation/reduction peaks of the CV curve is also affected by the diffusion rate of OH^−^ ions, manifesting as oxidation and reduction peaks moving in positive and negative directions, respectively. When the scanning rate is low, a large number of OH^−^ ions on the electrode surface participate in the Faradic redox reactions, that is, a quasi-equilibrium state. With the gradual increase in the scanning rate, the OH^−^ ions on the electrode surface drop sharply and those in the electrolyte cannot rapidly diffuse to the electrode surface, resulting in a disturbed equilibrium state. In conclusion, the position of the oxidation/reduction peaks and the surrounding area of the CV curve are affected by the diffusion rate of OH^−^ ions, thus confirming the pseudo-capacitance characteristics of the electrode material.

GCD tests were performed with a potential window ranging from 0 to 0.4 V at the current density of 2 A·g^−1^ to further explore the electrochemical performance of the electrode material (Figure 8a). The GCD curve of NF was approximately a straight vertical line, indicating that it virtually has no ability to store charge. This finding was consistent with the CV curve. In addition, the GCD curves of Ni_3_S_2_/NF and Ni(OH)_2_/Ni_3_S_2_/NF electrodes showed charge–discharge platforms related to redox reactions, and the discharge time of the latter (353.2 s) was longer than that of the former (93.2 s). This result indicated that the Ni(OH)_2_/Ni_3_S_2_/NF derived from Ni-MOF shows the highest capacitance. As calculated in Formula (2), the specific capacitances of NF, Ni_3_S_2_/NF, and Ni(OH)_2_/Ni_3_S_2_/NF were 2.6, 466.0, and 1766.0 F·g^−1^ at 2 A·g^−1^, respectively. The specific capacitance of Ni(OH)_2_/Ni_3_S_2_/NF was greatly improved at about 3.8 times that of Ni_3_S_2_/NF. GCD curves were obtained at different current densities ranging from 1 A·g^−1^ to 10 A·g^−1^ to further determine the specific capacitance of Ni(OH)_2_/Ni_3_S_2_/NF (Figure 8b). When the current density gradually increased, the Faradic redox reaction between the electrode and electrolyte interface decreased and the impedance value increased, resulting in an increase in voltage drop. From 1 A·g^−1^ to 10 A·g^−1^, the IR was calculated to be 0.002, 0.013, 0.029, 0.039 and 0.051 V. The relation of the specific capacitance of Ni(OH)_2_/Ni_3_S_2_/NF with different current densities is shown in Figure 8c. When the current density was 1, 2, 4, 6, 8, and 10 A·g^−1^, the specific capacitance was 2207, 1766, 1620, 1515, 1440, and 1375 F·g^−1^, respectively. When the current density was increased to 10 A·g^−1^, the specific capacitance remained at a relatively large value (62.3% of initial value), indicating the remarkable rate capability of Ni(OH)_2_/Ni_3_S_2_/NF. The specific capacitance decreased correspondingly with the increase in current density due to the insufficient utilization of active material caused by the rapid Faradic redox reactions at a high current density. According to the curves, the maximum specific capacitance of Ni(OH)_2_/Ni_3_S_2_/NF can reach 2207 F·g^−1^, which is significantly higher than those in the literature (Table 1). In addition, the GCD curves can reflect the rate capacity of electrode materials.

Figure 9 shows the capacitance retention and Coulombic efficiency of Ni(OH)_2_/Ni_3_S_2_/NF after 5000 cycles at a scanning rate of 50 mV·s^−1^. The Coulombic efficiency of Ni(OH)_2_/Ni_3_S_2_/NF was 81.4%. After 500 cycles of CV tests, the electrode was able to maintain 90.4% of its initial specific capacitance and showed virtually no change afterward. The reduction in the specific capacitance may be due to the active materials on the electrode surface slightly falling off from the current collectors during the early cycles, a common phenomenon caused by volume expansion [32]. After 500 cycles, the extremely high capacitance retention rate may be attributed to the unique morphology of the electrode. During this process, OH^−^ ions can completely penetrate the interior (Ni_3_S_2_) of the electrode material and undergo redox reactions. However, under the protection of the external material Ni(OH)_2_, the volume change of Ni_3_S_2_ will be limited, thus greatly reducing the shedding of Ni_3_S_2_. Meanwhile, the porous honeycomb structure can alleviate the volume stress of the material to a certain extent, thus leading to excellent cycling stability [28]. The above factors lead to the synergistic effect and promote the cycling stability of Ni(OH)_2_/Ni_3_S_2_/NF. Compared with those in the literature [46,47,48,49,50,51], the sample showed better cycling capability (Table 2).

Impedance affects the ability of an electrode material to store charge to some extent by hindering the transport of electrons/ions. EIS was tested in the frequency range of 10^5^ to 0.01 Hz to investigate the impedance of Ni_3_S_2_/NF and Ni(OH)_2_/Ni_3_S_2_/NF (Figure 10) (inset displays the fitting circuit). The components of the fitting circuit are closely related to the Nyquist plots, which are composed of a small semicircle in the high frequency region and a straight line in the low frequency. *Rs* is the equivalent series resistance (ESR), and its value corresponds to the intercept value on the real axis (Z’), reflecting the resistance of the electrolyte, electrode material inherent resistance, and active material/current collector interface contact resistance [20,26]. The smaller the *Rs*, the higher the conductivity and the more easily the electrons are transported. The *Rs* of Ni_3_S_2_/NF and Ni(OH)_2_/Ni_3_S_2_/NF were about 0.48 and 0.46 Ω, respectively, which may attributed to the binder-free structure without the obstruction of electron transportation. *Rct*, which stands for charge transfer resistance, is related to the diameter of the semicircle [52]. Its value reflects the difficulty level of the Faradic redox reactions. The lower the *Rct*, the easier the charge transfer and thus the Faradic redox reactions. Compared with that of Ni_3_S_2_/NF (0.56 Ω), the *Rct* of Ni(OH)_2_/Ni_3_S_2_/NF was significantly reduced (0.22 Ω). This result indicated that charges are easily transferred in the electrode material and electrolyte and further promote Faradic redox reactions. This phenomenon is due to the large specific surface area and high porosity of Ni-MOF-derived Ni(OH)_2_, which greatly reduces the transport path of active ions. Warburg impedance (*W_0_*), which reflects the diffusion rate of active ion inside the electrode materials, is related to the slope of a straight line in the low frequency region [11]. The higher the slope, the lower the diffusion resistance and the better the electrochemical performance of the electrode material. The slope of Ni(OH)_2_/Ni_3_S_2_/NF was greater than that of Ni_3_S_2_/NF, suggesting that the *W_0_* of Ni(OH)_2_/Ni_3_S_2_/NF (0.32 Ω) is smaller than Ni_3_S_2_/NF (0.65 Ω) due to the synergistic effect of a large contact area, a high specific area, and a high porosity. Meanwhile, a constant phase element was used to form a fitting circuit. The results demonstrated the advantage of low impedance of the Ni(OH)_2_/Ni_3_S_2_/NF electrode for an excellent electrochemical performance.

### 3.4. Electrochemical Measurements of the Ni(OH)_2_/Ni_3_S_2_/NF//AC HSC Device

A HSC device was assembled to further evaluate the practical application of the Ni(OH)_2_/Ni_3_S_2_/NF electrode (Figure 11). An Ni(OH)_2_/Ni_3_S_2_/NF electrode was used as the positive electrode, AC as the negative electrode, filter paper as the separator, and 6 M KOH as the electrolyte. As a key factor for evaluation, the mass ratio of Ni(OH)_2_/Ni_3_S_2_/NF and AC (apart from the NF current collector) was calculated using Formulas (1) and (3). Figure 12a shows the CV curves of Ni(OH)_2_/Ni_3_S_2_/NF and AC with voltage ranges of −0.1 V to 0.6 V and −1 V to 0 V, respectively, in a three-electrode system at a scanning rate of 50 mV·s^−1^. From the CV curves, the specific capacitances of Ni(OH)_2_/Ni_3_S_2_/NF and AC were calculated to be 1063.5 and 143.1 F·g^−1^, respectively (Formula 1). Therefore, the mass ratio of Ni(OH)_2_/Ni_3_S_2_/NF and AC was calculated to be approximately 0.2 by substituting the above values of the specific capacitance into Equation (3). Finally, the calculated masses of Ni(OH)_2_/Ni_3_S_2_/NF and AC were 4.6 and 23.0 mg, respectively, and both electrodes were assembled into the HSC device. The mass of the entire HSC device was the sum of the two active materials.

Figure 12b displays the CV curves of the device with the scanning rate ranging from 5 mV·s^−1^ to 100 mV·s^−1^ at a voltage window of 0–1.5 V. The shapes of all CV curves remained unchanged, indicating that 1.5 V is a suitable working potential range for the HSC device. GCD curves (Figure 12c) were also tested to identity the electrochemical performance of the HSC device. The curves almost maintained their symmetry at different current densities, demonstrating that the device has good electrochemical reversibility. As calculated in Equation (4), the specific capacitances of the device were 78.4, 50.3, 39.5, 29.9, 23.2, 22.4, and 18.7 F·g^−1^ at the current density of 0.5, 1, 2, 4, 6, 8, and 10 A·g^−1^, respectively. From 0.5 A·g^−1^ to 10 A·g^−1^, the decrease in specific capacity can be attributed to an incomplete reaction at a large current density. Ragone plots with the energy and power density of the HSC device were obtained at different current densities using Equations (5) and (6) (Figure 12d). The device showed a maximum energy density of 24.5 Wh·kg^−1^, accompanied by a power density of 375.0 W·kg^−1^. Its energy density gradually reduced to 4.2 Wh·kg^−1^ as the current density increased to 10 A·g^−1^, accompanied by an increase in power density to 7.6 KW·kg^−1^. This value is higher than previous results [53,54,55,56,57].

## 4. Conclusions

(1) A honeycomb-like Ni(OH)_2_/Ni_3_S_2_/NF grown on an NF was prepared successfully via the two-step hydrothermal approach, followed by an alkaline treatment.

(2) The MOF-derived Ni(OH)_2_ nanosheets were grown on the Ni_3_S_2_/NF surface, resulting in the formation of a honeycomb-like Ni(OH)_2_/Ni_3_S_2_/NF that has a large specific area and porosity and can increase the active sites and shorten the ion transport pathway.

(3) The as-prepared Ni(OH)_2_/Ni_3_S_2_/NF electrode showed an outstanding specific capacitance (2207 F·g^−1^ at 1 A·g^−1^, 1929.7 F·g^−1^ at 5 mV·s^−1^) and retained 62.3% of the initial value at 10 A·g^−1^. Moreover, 90.4% of its initial specific capacitance was retained after 5000 cycles at 50 mV·s^−1^ in a three-electrode system.

(4) A HSC was assembled and tested in a two-electrode system with an Ni(OH)_2_/Ni_3_S_2_/NF electrode as the positive electrode, AC as the negative electrode, filter paper as the separator, and 6 M KOH as the electrolyte. The device displayed a superior electrochemical performance with a high energy density (24.5 Wh·kg^−1^ at a power density of 375 W·kg^−1^).

All the results demonstrated that honeycomb-like Ni(OH)_2_/Ni_3_S_2_/NF has great potential for application in high-performance supercapacitors. However, given that the electrode material determines the type of electrolyte, the working voltage of the aqueous electrolyte (KOH) will be limited by the voltage decomposition of water. According to Equations (5) and (6), despite the high specific capacity of this electrode material, its energy density is extremely low and greatly restricts its commercialization. Appropriate composites of carbon and MOF-derived Ni(OH)_2_ can be compounded on the surface of Ni_3_S_2_/NF to improve the electrochemical performance of Ni(OH)_2_/Ni_3_S_2_/NF. In addition, an appropriate amount of active material must be attached to Ni_3_S_2_/NF by controlling the time or concentration of the hydrothermal process. If the attached active materials are excessive, then Ni_3_S_2_/NF cannot fully react with the ions in the electrolyte and the impedance will increase. As a result, the electrode material will have a low specific capacity and a poor rate capability and cycling stability. On the contrary, if the attached active materials are not enough to completely cover the underlying materials, then the active sites will be significantly reduced and the electrochemical performance will be poor. Carbon materials (graphene and carbon nanotube) with excellent electrical conductivity, a large specific surface area, and high mechanical flexibility can promote the ion/electron transport rate while further reducing the volume change caused by OH¯ intercalation/deintercalation in Ni(OH)_2_/Ni_3_S_2_/NF, thus providing the electrode material with a high specific capacity and an excellent rate capability and cycling stability.

## Figures and Tables

**Figure 1 nanomaterials-13-00493-f001:**
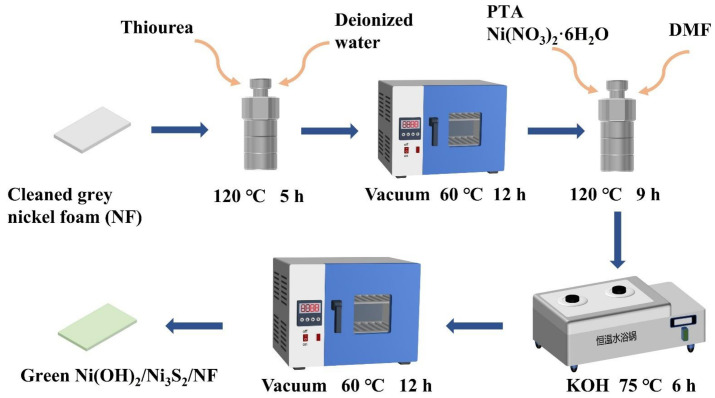
Schematic illustration of the preparation process for green Ni(OH)_2_/Ni_3_S_2_/NF.

**Figure 2 nanomaterials-13-00493-f002:**
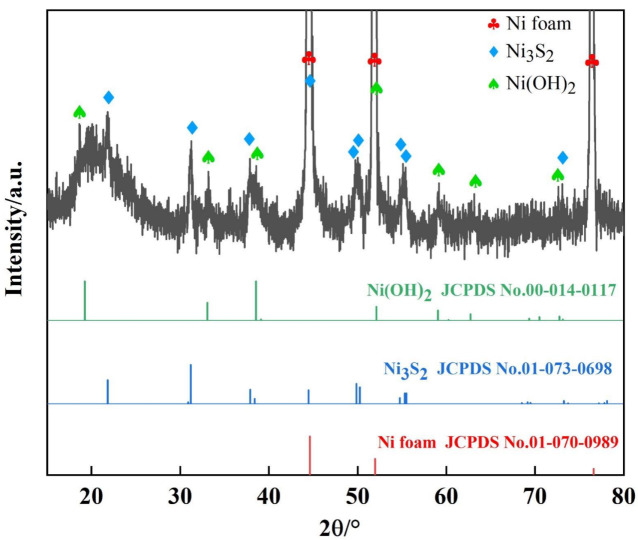
XRD patterns of the Ni(OH)_2_/Ni_3_S_2_/NF electrode.

**Figure 3 nanomaterials-13-00493-f003:**
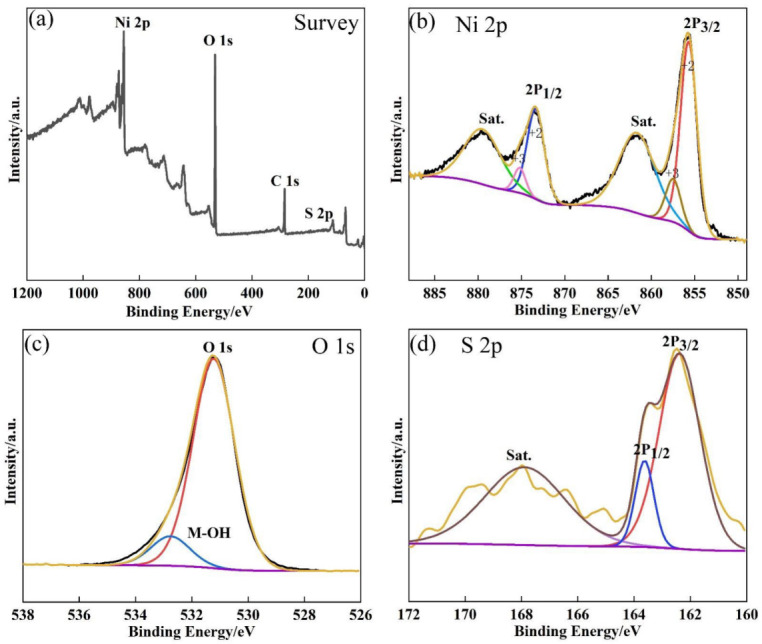
XPS spectra of the Ni(OH)_2_/Ni_3_S_2_/NF electrode. (**a**) Survey; (**b**) Ni 2p; (**c**) O 1s and (**d**) S 2p.

**Figure 4 nanomaterials-13-00493-f004:**
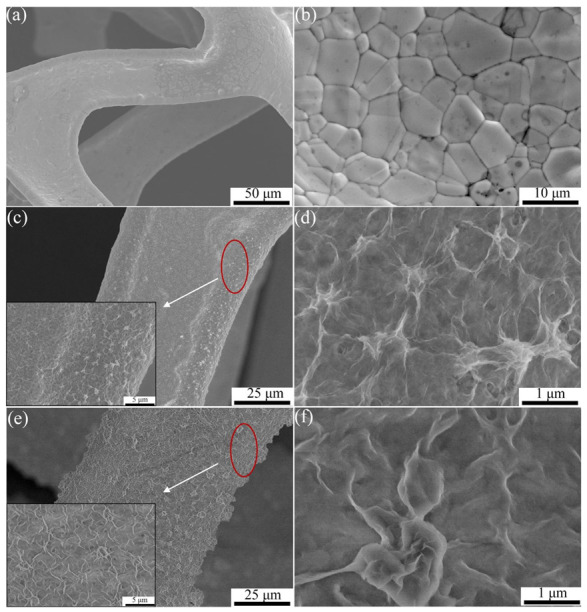
FE-SEM images of the electrodes. (**a**,**b**) pure NF; (**c**,**d**) Ni_3_S_2_/NF and **(e**,**f)** Ni(OH)_2_/Ni_3_S_2_/NF.

**Figure 5 nanomaterials-13-00493-f005:**
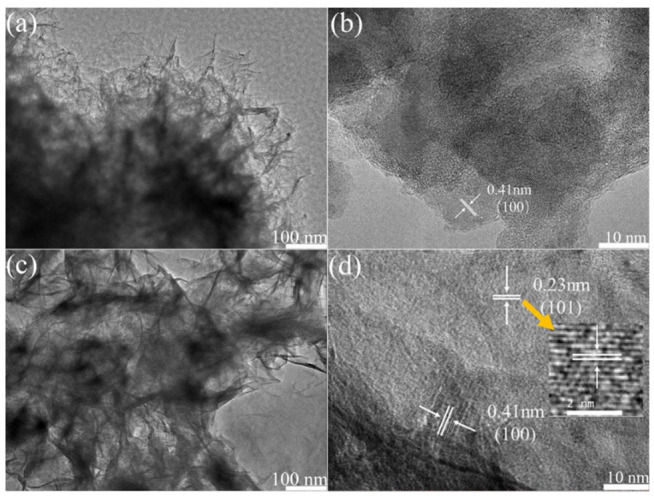
TEM and HRTEM images of (**a**,**b**) Ni_3_S_2_/NF; (**c**,**d**) Ni(OH)_2_/Ni_3_S_2_/NF.

**Figure 6 nanomaterials-13-00493-f006:**
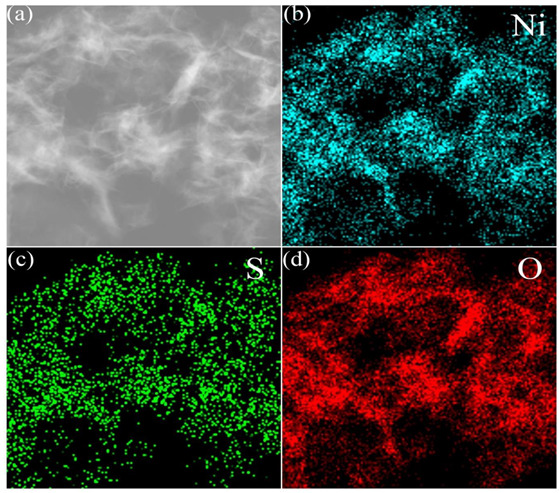
(**a**–**d**) EDS mapping of Ni, S and O elements in Ni(OH)_2_/Ni_3_S_2_/NF.

**Figure 7 nanomaterials-13-00493-f007:**
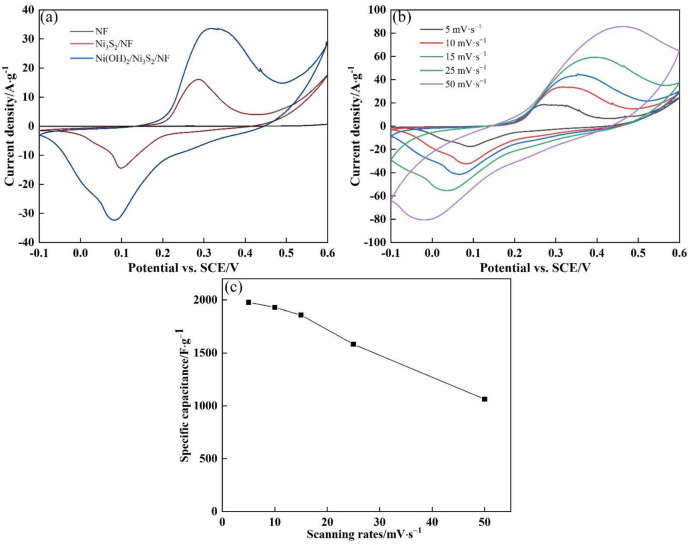
(**a**) CV curves of NF, Ni_3_S_2_/NF and Ni(OH)_2_/Ni_3_S_2_/NF at the scanning rate of 10 mV·s^−1^; (**b**) CV curves of Ni(OH)_2_/Ni_3_S_2_/NF at the scanning rate ranging from 5 mV·s^−1^ to 50 mV·s^−1^; (**c**) Specific capacitance of Ni(OH)_2_/Ni_3_S_2_/NF at different scanning rates.

**Figure 8 nanomaterials-13-00493-f008:**
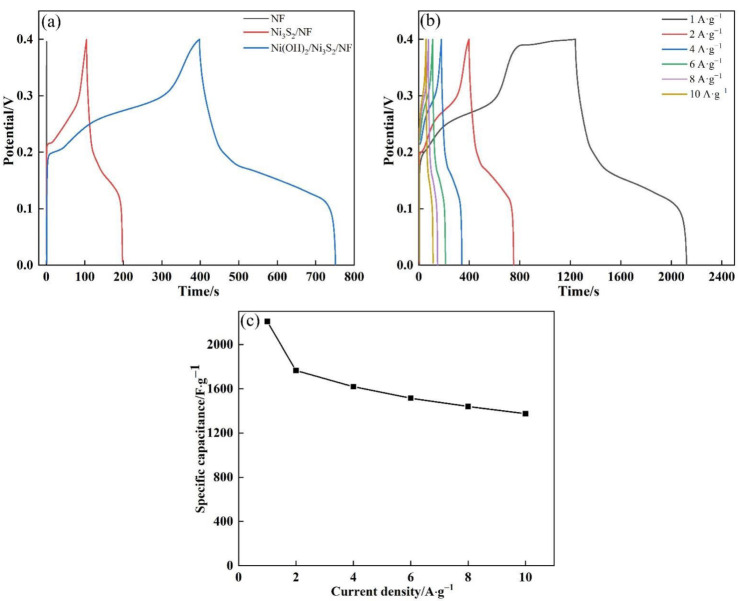
(**a**) GCD curves of NF, Ni_3_S_2_/NF and Ni(OH)_2_/Ni_3_S_2_/NF at the current density of 2 A·g^−1^; (**b**) GCD curves of Ni(OH)_2_/Ni_3_S_2_/NF at the current density from 1 to 10 A·g^−1^; (**c**) Specific capacitance of Ni(OH)_2_/Ni_3_S_2_/NF at different current densities.

**Figure 9 nanomaterials-13-00493-f009:**
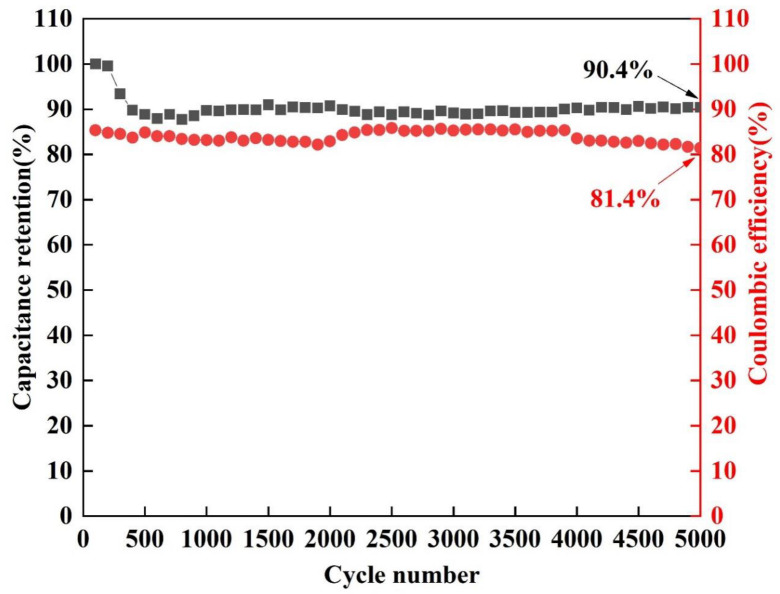
Cycling tests for Ni(OH)_2_/Ni_3_S_2_/NF at the scanning rate of 50 mV·s^−1^ up to 5000 cycles.

**Figure 10 nanomaterials-13-00493-f010:**
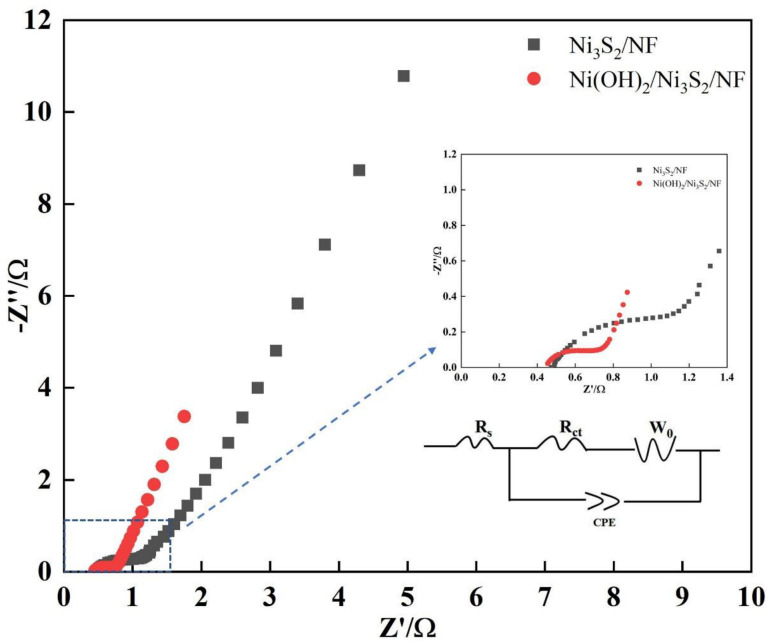
Nyquist plots of Ni_3_S_2_/NF and Ni(OH)_2_/Ni_3_S_2_/NF.

**Figure 11 nanomaterials-13-00493-f011:**
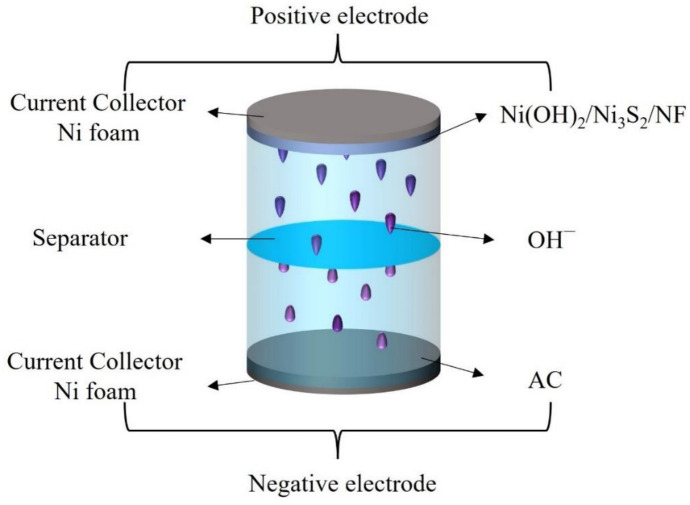
Schematic diagram of the Ni(OH)_2_/Ni_3_S_2_/NF//AC hybrid supercapacitor (HSC).

**Figure 12 nanomaterials-13-00493-f012:**
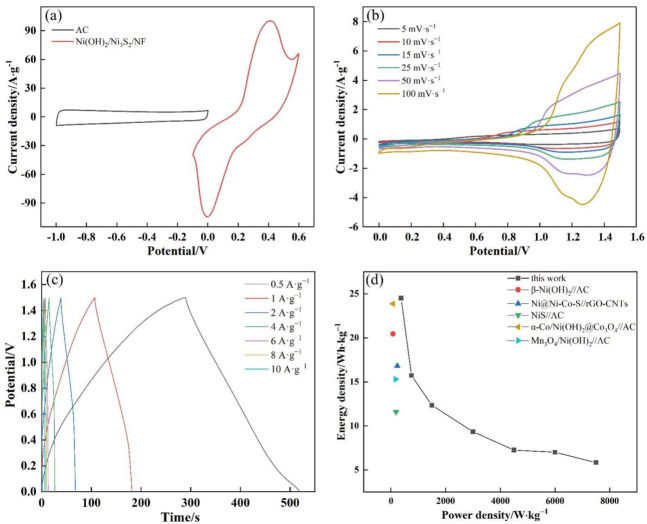
(**a**) CV curves of the AC and Ni(OH)_2_/Ni_3_S_2_/NF electrode at 50 mV·s^−1^; (**b**) CV curves of the HSC device at different scanning rate ranges from 5 to 100 mV·s^−1^; (**c**) GCD curves of the HSC device at different current density from 0.5 to 10 A·g^−1^; (**d**) Ragone plot of the Ni(OH)_2_/Ni_3_S_2_/NF//AC HSC.

**Table 1 nanomaterials-13-00493-t001:** Comparison of electrochemical performance of Ni(OH)_2_/Ni_3_S_2_/NF with those reported in the other related literatures.

Refs.	Materials	Electrolyte	Performance	Cyclic stability
This work	Ni(OH)_2_/Ni_3_S_2_/NF	6 M KOH	2207 F·g^−1^, 1 A·g^−1^	5000, 90.4%
[24]	Ni_3_S_2_/Ni(OH)_2_	3 M KOH	2065 F·g^−1^, 1 A·g^−1^	3000, 82.1%
[35]	MoS_2_/Ni(OH)_2_	1 M KOH	2192 F·g^−1^, 1 A·g^−1^	10,000, 90.6%
[43]	Ni_3_S_2_	PVA/KOH	981.8 F·g^−1^, 2 A·g^−1^	1000, 96.9%
[44]	Co_9_S_8_/Ni_3_S_2_/NF	2 M KOH	1356 F·g^−1^, 1 A·g^−1^	2500, 80.8%
[45]	NiCo_2_O_4_/Ni(OH)_2_	1 M KOH	1336 F·g^−1^, 1 A·g^−1^	/

**Table 2 nanomaterials-13-00493-t002:** Comparison of capacitance retention of electrode materials with those reported in the other related literatures after different cycle number of charge–discharge tests.

Refs.	Materials	Cycle number	Capacitance retention
This work	Ni(OH)_2_/Ni_3_S_2_/NF	5000	90.4%
[46]	Co_9_S_8_@Ni_3_S_2_/ZnS	4000	79.7%
[47]	C/NiMn-LDH/Ni_3_S_2_	5000	84.25%
[48]	Ni_3_S_2_/NiO	4000	90.2%
[49]	Ni(OH)_2_/NF	3000	90%
[50]	Co_3_O_4_/Ni(OH)_2_	5000	80.1%
[51]	NiCo_2_O_4_-C@Ni(OH)_2_	5000	79%

## Data Availability

Not applicable.

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
