# Peer review of "Synthesis of Ni3S2 and MOF-Derived Ni(OH)2 Composite Electrode Materials on Ni Foam for High-Performance Supercapacitors"

_nanomaterials, 2023, doi:10.3390/nano13030493_

Round 1

Reviewer 1 Report

The specific manuscript has been reviewed. While it is not a breakthrough research topic, the overall quality of the manuscript is in a standard to be published. The presentation of the manuscript is in general in a good shape.

some improvements can be made:

1: There have been large amount of similar material reported with similar energy density. What is the gap for such material to be commercialized? The authors can discuss it.

2: The author can discuss the future improvement of the material.

Reviewer 2 Report

The authors provided an engaging and easy-to-synthesize honeycomb-like Ni(OH)2/Ni3S2/Ni foam material that presented remarkably high specific capacitance, rate capability, and cycling stability. Also, the hybrid supercapacitor exhibited an excellent energy density. They provided complete physical-chemical characterizations, including microscopy, XPS, and XRD, which is necessary to know the system better. However, as a minor revision, the authors should correlate the physical characterizations with the electrochemical ones. For me, it seemed that the physical characterizations were there without function once they did not use them to explain their findings. 

Reviewer 3 Report

Recommendation: Publish after minor revision noted.

In this work, the author reported “Synthesis of Ni3S2 and MOF-derived Ni(OH)2 composite electrode materials on Ni foam for high-performance supercapacitors and systematically characterized using various physiochemical techniques. I strongly believe that the research part has been experimented in a proper procedure. Therefore, I recommended this work for publication in the Nanomaterials journal. However, some of the minor concerns should be addressed before proceeding with further actions.

1.      Use the proper keywords which explain the significance of the work.

2.      Fig. 7 a and b the X-axis title must be Potential vs. Ag/AgCl (V).

3.      Authors should calculate the IR drop in Fig.8b

4.      The table 1 should have cyclic stability.

5.      Authors should calculate ESR value of nyquist plot.

6.      Authors should include coulombic efficiency along with capacitance retention.
